# Wild Plants Potentially Used in Human Food in the Protected Area "Sierra Grande de Hornachos" of Extremadura (Spain)

**José Blanco-Salas \***[ID]**, Lorena Gutiérrez-García**[ID]**, Juana Labrador-Moreno**[ID] **and Trinidad Ruiz-Téllez**[ID]

Department of Vegetal Biology, Ecology and Earth Science, University of Extremadura, 06071 Badajoz, Spain; zoolorena5@hotmail.com (L.G.-G.); labrador@unex.es (J.L.-M.); truiz@unex.es (T.R.-T.)
**\*** Correspondence: blanco_salas@unex.es; Tel.: +34-924-289-300 (ext. 89052)

**Abstract:** Natura 2000 is a network of protected spaces where the use of natural resources is regulated through the Habitat Directive of the European Union. It is essential for the conservation of biodiversity in Europe, but its social perception must be improved. We present this work as a demonstration case of the potentialities of one of these protected areas in the southwest (SW) Iberian Peninsula. We show an overview of the catalog of native wild plants of the place, which have nutritional and edible properties, having been used in human food by the peasant local population over the last century, and whose consumption trend is being implemented in Europe mainly through the haute cuisine and ecotourism sectors. What is offered here is a study of the case of what kind of positive contribution systematized botanical or ethnobotanical scientific knowledge can make toward encouraging innovative and sustainable rural development initiatives. A total of 145 wild plants that are potentially useful for leading tourism and consumers toward haute cuisine, new gastronomy, enviromentally-friendly recipes, and Natura 2000 Conservation are retrieved. The methodology used for our proposal is based on recent proposals of food product development and Basque Culinary Center initiatives.

**Keywords:** local foods; bioactive components; traditional recipes; traditional dietary patterns; edible plants; Mediterranean; innovative gastronomy

---

## 1. Introduction

### 1.1. Plant Resources Global Availability and Potential Use

From approximately 250,000 plant species, it has been estimated that up to 75,000 could be edible [1], and some 7000 are regularly eaten worldwide [2]. Even though the average Western citizen has now access to more species of edible plants than ever, a common household shopping list will not include more than 45 plant species as food supplies. In fact, Western societies tend to use much fewer species than indigenous communities. Regarding wild resources (plant species that grow spontaneously in populations in natural or managed habitats, thriving independently of direct human action, following the definition of [3], 73 plant species are reported to be used in Álava (Spain, [4]), while 66 plant species are reported to be used in Emilia (Italy, [5]), the inhabitants of the Quinling Mountains of Saanxi (China) use up to 185 species [6], and the indigenous peoples of southern Ecuador use up to 354 [7]. The main actor responsible for this steady loss of culinary traditions is globalization [8]; wild edible plants are also often stigmatized due to associations with food shortage and poverty, climatic extreme events, or politically conflictive times ([9] and references therein), leading to the abandonment of wild edible resources and the consequent decline of knowledge

in plant uses. What has become increasingly clear is that any approach to biodiversity and landscape conservation must incorporate traditional knowledge in order to achieve success.

*1.2. Combined Conservation Trends Focusing on the Natura 2000 Network*

In rural communities of southern Spain, a close relationship between biodiversity, local culture, and dialectal diversity has evolved over centuries of history. There is a genuine biocultural heritage [10] that has been specifically developed on a local spatial scale and has been until recent times transmitted through generations. There, significant areas belong to the Natura 2000 Network. This is the organized area for the conservation of biodiversity in Europe. It is regulated through the Habitats Directive of the European Union, which enables compatible development, knowledge, bioeconomy, and innovation [11]. However, sometimes, the measures proposed by the environmental organisms are seen by the local population with suspicion or rejection, and they are challenged [12]. The problem has worried the European authorities to the point of having implemented specific action programs aimed at improving its social perception [13]. Most are oriented to new trends and development opportunities to favor biodiversity and Natura 2000 Network acceptance.

*1.3. The Particular Case of the Sierra Grande de Hornachos: Ecological and Cultural Relevance*

Hornachos is a small town (3777 inhabitants) [14] of the central Badajoz province (Extremadura) in the southwest of the Iberian Peninsula that has this problematic. It provides an optimal study case for biocultural heritage whose record and divulgation is timely. In addition to this, there is a sociological reality in the territory of unpopularity related to the Natura 2000 Network. In this framework, we are actually carrying out a project supported by the European Union and the regional government. It is intended in the medium term to solve real problems of rejection and blockage that have occurred in the past. An SAC and SPA (Special Area for Conservation and Special Protection Area for Birds) have been developed in the "Sierra Grande de Hornachos" within the Natura 2000 Network [15]. Its forests and thickets represent an ideal environment for the life of abundant Mediterranean wild fauna of high value with elements such as the griffon vulture, the golden eagle, and more than 220 species of vertebrates. There is the cartography of the predominance of holm oaks in flat areas, hillsides, and foothills; of cork oaks in shady and watercourses; of mixed forest; of wild olive trees; of summit junipers; and of *Nerium oleander (adelfares)* and *Flueggea tinctoria (tamujares)* in the riparian vegetation. Surprisingly, the flora has been much less inventaried, and no specific floristic catalog of the area has been published to date. For ethnobotanical studies, there are also no specific systematized works, despite the potential of the place. It has a rich historical past that includes a mixture of cultures, such as Christian, Jewish, and Muslim. This circumstance has produced a strong imprint in the agriculture of the area. It influenced the maintenance of agroecosystems, including many activities related to nature, hunting, fishing, medicine, gastronomy, and even traditional crafts, which are as interesting as those from Andalucía [16].

*1.4. Local Traditional Knowledge on Food Plants and Resource Management at Sierra Grande de Hornachos*

In this region, as well as in others of the Mediterrenean [17], economic marginality, historical factors, and recurrent political crises in old times forced its inhabitants to make the most out of the wild resources found in the vicinity of their settlement. However, over the last decades, the development of agriculture and the globalization phenomena have led to the progressive abandonment of wild edible resources [9,17], and nowadays, it is still a rich bio-heritage that holds value.

Taking into consideration that the regional flora of Extremadura has 1938 species [18] and that the "Sierra Grande de Hornachos" comprises 12,469 ha of well-conserved Mediterranean vegetation units, its floristic resources were expected to be interesting. In a first aproximation, a list that did not reach 50 species was the only published result [19], featuring threatened plants such as *Lavatera triloba*, *Erodium mouretii*, *Orchis italica* and *Ophrys fusca* subsp. *dyris* as standouts. The complete floristic inventory of the area was approached by our team in 2016/17 Project IB16003 with the expectation

of finding at least 400 species. We made field prospections, and to date, 1301 taxa make up the checklist of the wild vascular plants of Sierra Grande de Hornachos, according to our campaigns and investigations. This systematical biodiversity information (of 1301 taxa) can be crossed with another source (database) that is not open access, but is available in the Ministry of the Environment of Spain: The Spanish Inventory of Traditional Knowledge Relating to Biodiversity. Doing this, all of the vegetal resources can be studied and presented to the local inhabitants of the region as possibilities for the development and progress toward a sustainable society of 21st century. The real challenge relating to the conservation of rural areas and phytodiversity in Europe is to find new trends and opportunities for plants and people. That seems to be the most sucessful way to provide social support for the Natura 2000 Network. This is the main focus of our work. Constructing the catalog of the wild plants of the area, we can look for the uses of these species and explore their possibilities. We can study how they were employed in other places in the country. This means that they can effectively be also utilized in the same way. Thus, the traditional used and conserved knowledge will drive innovative applications in the area. In summary, a positive view of the sustainable exploitation of natural resources in the Natura 2000 Network can promote a friendly face of conservation measures. For this reason, it is important to explore this method of management.

### 1.5. Innovation for Conservation: Wild Gathering, Agro-Tourism, Gourmet Foods, and Local Memories

New initiatives are starting in Europe. Agro-tourism is rapidly increasing its popularity through urban Western societies, ultimately enabling the survival of traditional knowledge and the gathering of wild plants itself [20]. Nevertheless, a renaissance of interest in traditional plant knowledge is beginning. Gathering wild edible plants has become popular as a means of enjoying outdoor contact with nature and granting alternative, high-end culinary experiences [9]. Wild plant consumption is not unfainly seen as "an occupation for the poor, which increased during times of bad crops and famine", but also a fashionable part of 21st century haute cuisine [17]. It is even increasingly seen as a form of reaffirming cultural identity, sophistication, and commitment to the community. In brief, innovative strategies suggested for natural resources are excellent tools for promoting the culture of biological conservation among social sectors, and hopefully serve to mitigate local rejects to Natura 2000 regulations.

In that framework and with the underlying characteristics of the case of Hornachos, we state the specific objectives for this work.

### 1.6. Objectives

The objective of this work is to record the threatened biocultural heritage of a Natura 2000 area (Hornachos) on wild edible plants with the double purpose of preserving it and setting the basis for a valorization and divulgation program focused on the local and visiting population.

## 2. Materials and Methods

1. The database of the Spanish Inventory of Traditional Knowledge Relating to Biodiversity (IECTB) [21–24] comprising of 3156 record (taxa) was taken as a basis. To date, it is not yet open access. So, permission to consult it was formally required by the authors (J. Blanco-Salas) and allowed by the IECTB national coordinator (M. Pardo de Santayana).

   - IECTB is the Inventory of Traditional Knowledge that has been published in Spain but dispersed in a difficult literature. The realization of this inventory arose from a legislative requirement (Article 70 of Law 42/2007, of December 13, the Natural Heritage and Biodiversity of Spain). In this inventory, the preparation of plants for food uses is written in the same way as it has been reported from local informants.

2.  In parallel, our research group created a database of Hornachos plants. Basic information was taken from the Global Biodiversity Information Facility (GBIF) [25], the open-access program ANTHOS [26], and the Herbarium UNEX (University of Extremadura) collections.

    - GBIF [25] stands for the Global Biodiversity Information Facility. It is an international network and research infrastructure funded by the world's governments that is aimed at providing anyone, anywhere, open access to data about all types of life on Earth.
    - ANTHOS [26] is an open-access program that was developed to divulge and show society information about the biodiversity of Spanish plants on the Internet. The initiative was born under the umbrella of the Flora Iberica research project (Ministry of Agriculture, Food, and the Environment) and the Royal Botanical Garden of Madrid, Spain.
    - UNEX Herbarium is the herbarium of the University of Extremadura, which was formed by 36,451 specimens of vascular plants whose main origin is the autonomous region of Extremadura (Spain) and Portugal. Here is the place where we included the voucher speciments of our field collections.

3.  The Hornachos list finally resulted in 1301 plants. We looked for 1301 names in the IECTB Inventory official database, and 834 matched (Figure 1). This means that we firstly had 834 useful taxa and 1301−834 = 467 taxa without use.

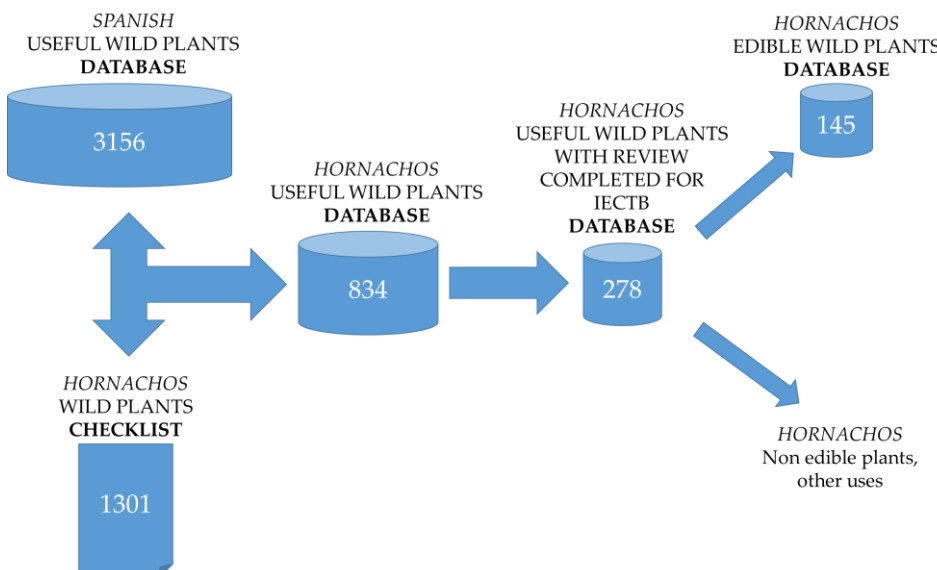

**Figure 1.** Methodology scheme.

4.  The IECTB inventory database information compiled in "one file for plant" format has not yet been completely published. Of the 834 useful Hornachos plants included in the IECTB, to date, the IECTB has made information on 278 species available in published books [21–24] (the remainder of the 834 species will be published in the future, when the IECTB project is finished). Of these 278 useful Hornachos plants, 145 species were edible plants. These names of edible plants were listed.

5.   A provisional database of the edible plants of Hornachos was built, with 145 records (corresponding to the provisional 145 edible plants in Hornachos).

- Botanical nomenclature was adapted to the Angiosperm Phylogeny Group (APG) IV System [27] and included vernacular names.
- Part of the plant used (upon the information taken from the IECTB).
- Subcategory of food use. The ones considered were: A = Vegetables; B = Roots, bulbs, tubers, and rhizomes; C = Sweet fruits; D = Dry and oleaginous fruits; E = Cereals and pseudocereals; F = Fat; G = Alcoholic drinks; H = Non-alcoholic drinks; I = Condiments; J = Sugars and sweeteners; K = Candies and chewing; L = Other food uses. They are the same as considered by the IECTB. The information for filling the corresponding fields was literally transferred from the IECTB.
- Preparation. It was filled upon the following classification: 1 = Soups; 2 = "gofio" (= roasted cereal stirred into liquid); 3 = Broths and purees; 4 = Omelettes; 5 = Cakes; 6 = Rices; 7 = Salads; 8 = Cold vegetable soups; 9 = Pies/Patties; 10 = Potages; 11 = Stews; 12 = Scrambled; 13 = Sautéed/boiled/toasted; 14 = Milky; 15 = Fried/breaded; 16 = Sauces; 17 = Oil; 18 = Vinegar; 19 = Renner; 20 = Preservative; 21 = Dressings/condiments; 22 = Flour; 23 = Sugar; 24 = Desserts; 25 = Ice creams; 26 = Sorbets; 27 = Cakes/Cakes; 28 = Sweet; 29 = Jams; 30 = Marmalades; 31 = Jellies; 32 = To the natural; 33 = Brine; 34 = Pickles; 35 = In syrup; 36 = In alcohol; 37 = Wines; 38 = Liquors; 39 = Syrups for cocktails; 40 = Juices; 41 = Brandies; 42 = Infusions; 43 = Coffee/tea; 44 = Diluted syrups. This classification is based on that of Bertrand [28]. We assigned the information about preparation, which was recorded in IECTB format, to the corresponding culinary categories of Bertrand.

6.   Finally, tables and graphs were made in order to facilitate discussing the results.

## 3. Results

The constructed database had a total of 145 records. For each record, we considered the nomenclatural field (family, scientific, and vernacular), 15 sub-categories of food uses according to the IECTB, 42 Bertrand food classifications, and 22 possible parts of the the plant used. The total matrix of managed results included $145 \times 3 \times 15 \times 42 \times 22 = 6{,}029{,}100$ possible raw data. The raw data results are summarized in Table A1 of Appendix A, and the corresponding boxes of Appendix B (Boxes A1–A8). The resulting 145 species belong to 49 families. Asteraceae is the best represented, with 25 different vegetables, and Lamiaceae stands out for the preparation of alcoholic and non-alcoholic drinks, as well as condiments. Figure 2 summarizes the diversity of uses among the taxonomic groups' plant uses, which covered up to 12 subcategories of the IECTB, mostly involving brewing and the aromatization of beverages (alcoholic, 51 species, and non-alcoholic, 28 species). Up to 39 species were used as a form of "wild candy", being picked and chewed as mere entertainment. In the following paragraphs, we discuss the scope of the applications found.

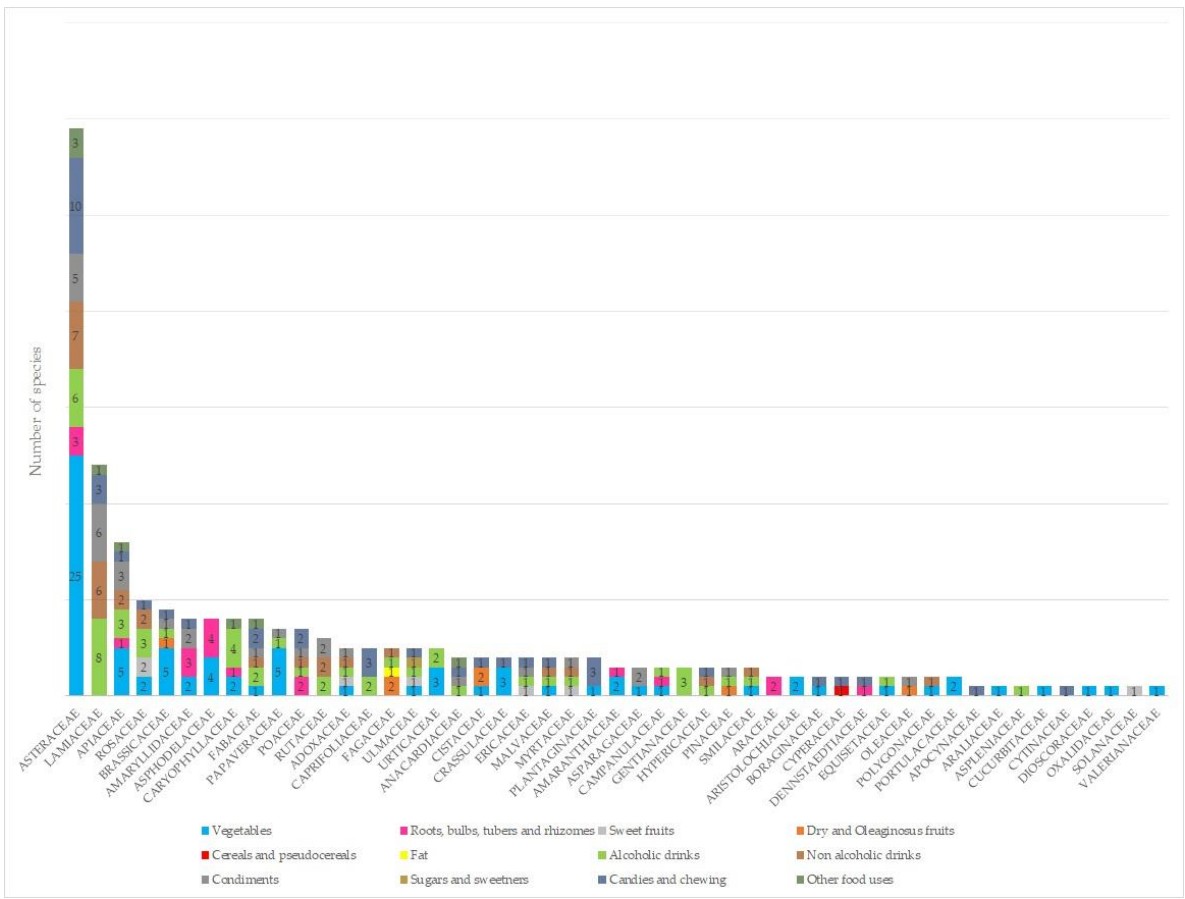

**Figure 2.** In colors, the diversity of uses among the families. Each bar shows its corresponding number of species.

## 4. Discussion

### 4.1. Hornachos Natura 2000 Area and Wild Edible Plants: Diversity and General Possibilities

The diversity of edible plants in the Hornachos community catalogued to date is high in comparison with other territories of Europe. It is more than double the number of edible plants used in similar but more urbane environments of the Basque Country (northern Spain [4]) or Emilia (central Italy [5]) and is approaching figures that have been published for human settlements in rural Asia [6]. However, it is still far from those that have been published for the indigenous communities of South America [7]. Apparently, the rich biocultural heritage of Hornachos is still well preserved. This can be explained by the overall lower level of industrialization of Extremadura in comparison with the more developed areas of the country, where contact with nature is much lower, causing the progressive demise of traditional knowledge.

Asteraceae is the most useful family of our catalogue, but this is also one of the three floristically richest families of the regional flora [18]. As in other Mediterranean countries (see a compilation in [9]), the diversity of wild greens correlates with the presence of bitter, pungent, or acrid secondary metabolites that allow for a variety of nuances in the perception of taste. Another nutritional aspect that has been studied recently is the influence of micronutrients in the correct performance of many metabolic processes, even at the prenatal stage [29]. Experimental studies on women have demonstrated the influence and significance of a wild vegetable diet in folate levels, and similar nutritional parameters [30] and specific reviews have focused on the adolescents of developed and underdeveloped countries. [31] Traditional indigenous foods have been studied to evaluate their levels of micronutrients [32], and the suitability of wild vegetables for alleviating human dietary deficiencies

has been addressed [33]. Specific studies of plants that have been traditionally consumed in Spain [34] have also been made.

In addition to this, previous studies in the field of haute cuisine have been published on the possibilities of innovation in this field [35–38]. In Northern Europe [39,40], interesting new design initiatives have been developed related to high-level gastronomy [41–43]. Important professionals of the sector with recognized international prestige have proposed cooking with wild plants [28]. The avant-garde cuisine has been linked to the world of art through the creative universe of great chefs from southern Europe such as Ferrán Adriá, El Bulli, and Mugaritz. Specialized official universities have also become subjects, such as the Basque Culinary Centre in Mondragón (Spain) [44]. Following the artistic concept of Chinese cuisine [45], the model has become directed toward the elaboration of dishes [46,47] that are a mixture of art and science. Wild biodiversity has started to be considered as an interesting resource for the development of new gastronomic and edible products [48,49]. Ethnobotany and the study and cataloguing of traditional knowledge by the local population constitute an important resource for implementing new food product development strategies [50].

Figure 3 shows the number of species of our study, which can be organized according to the different culinary uses of Bertrand classification [28], based on the raw data compiled in Table A1. Apart from the individual potential of each species, we consider that the examples that we expose next have an added value.

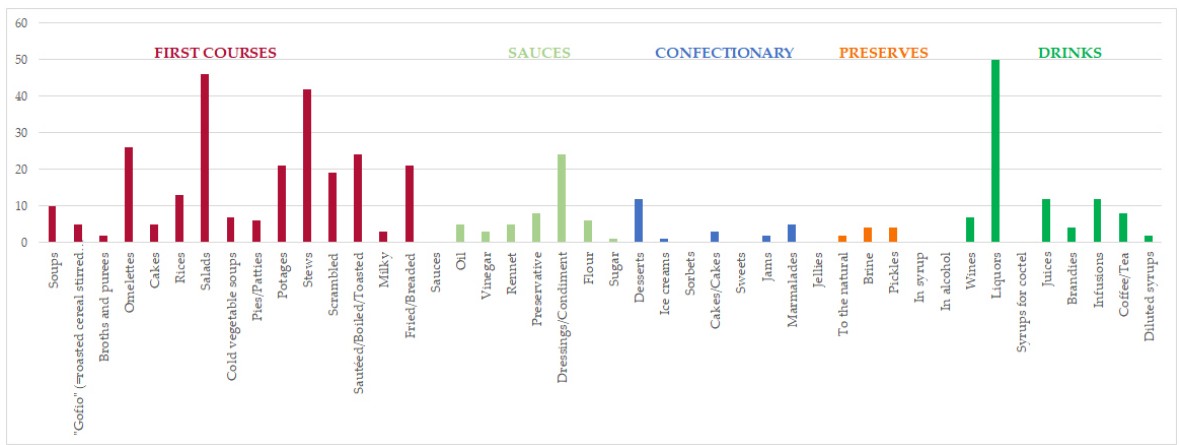

**Figure 3.** Distribution of the 145 species of the study according to the culinary classification proposed by Bertrand [28].

*4.2. Wild Edible Plants: Innovative Gastronomy and Healthy Promotion*

First courses and dishes can be creatively improved by the incorporation of many of the above-mentioned examples. Table A1 summarizes which vegetables have been experimented by traditional cooks, to prepare all kind of soups, broths, purees, omelettes, pies, patties, and salt cakes. The most common way of using the plants of our catalogue has been in stewing rices or potages, or adding them to salads. There is an immense arsenal of creativity at the disposal of the talent of the future professionals of haute cuisine. In Table A1, the species assigned to the Bertrand classification numbers 7, 10, and 11 are direct examples (last column). Another interesting new way of exploring haute cuisine is the new formulation of fried, breaded, sautéed, boiled, or toasted culinary preparations made out of the plants that we have assigned in Table A1 as belonging to the Bertrand classification numbers 13 and 15.

Soups can be made with *Chenopodium* or *Urtica*, which also carry *Allium* or *Foeniculum,* for example. The first one provides minerals, fiber, vitamins, and essential fatty acids that enhance the sensory and functional value of the food [51]. The latter is especially significant in the last two cases. Omelettes recipes can be based on *Crepis vesicaria* subsp. *haenseleri*, *Scorzonera laciniata*, *Sonchus oleraceus*, *Borago officinalis,* or *Eruca vesicaria*. There is even a tradition of using the young shoots of *Tamus communis*

and *Brionia dioica* as a substitute for *Asparagus* for this same purpose. Their particularly bitter taste makes them peculiar, and precaution about the preparation must be taken in account in order to avoid the unwanted effects that the bibliography describes [17]. Scrambled *Capsella bursa-pastoris* and cold vegetable soups (gazpachos) where *Sisymbrium orientale* or *Allium ampeloprasum* are added have a bromatological explanation. These plants are rich in sulfur components belonging to the glucosinolate group (for the *Brassicaceae*) and the non-proteic amino acids (for the species of *Allium*).

In any case, they are very odorous molecules with a high biological activity [52]. In a similar direction, the use of other species of *Allium* or elements of the mentioned familiy (e.g., *Rapistrum rugosum*, *Sisymbrium irio*), as proposed in Table A1, for the recipes for stew presentations, is worthwhile because it enables the incorporation of the phyochemical variability of the systematics to the cuisine. For the preparation of rice, *Sisymbrium orientale* is recommended. The same can be said about *Sonchus asper*, which has an excelent nutritional profile, being very rich in fiber and ω-3 acids [53]. We emphasize as well the singularity of the proposal of using previously cooked *Cytinus hypocistis* because it is a healthy and antioxidant food, but it is also very astringent [54]. Salad is a very fashionable group for gourmets in agro-tourism. New developments can be made by taking any of the plants that are codified with the number seven from from Table A1. Some of them (e.g., *Apium*, *Chenopodium*, *Chondrilla juncea*, *Bellis perennis*, *Montia fontana*, *Silene vulgaris*, *Stellaria media*, *Sonchus tenerrimus*, *Tragopogon porrifolius*, *Umbilicus rupestris*, *Valerianella microcarpa*, and *Veronica anagallis-aquatica*) are already entering the high Spanish hotel industry and occupying small selectede market niches. They have organoleptic and nutritional values [54] mainly as antioxidants [55], which is very interesting.

Another group that is worthy of mention for its variability is the drinks category. We have found a large list of plants that remain at the moment underexploited in the Hornachos local area, having a great potential in the industry of liquors, spirits, and soft drinks. Table A1 summarizes the potentialities of the flora of this Natura 2000 area in this sector (Bertrand classification numbers 36 to 44). Our proposal can really be transferred to land, because Hornachos is situated next to Tierra de Barros (Badajoz), which is an active vinegrowing district, with a developed liqueur industry. There is a possibility of making vinegar with rosemary oil and brandy with a few plants that could be explored. They are *Sambucus nigra*, *Foeniculum vulgare,* and *Arbutus unedo*, respectively, which are very rich [56–58] in phenolics derivatives. Diluted syrups with *Achillea ageratum* and *Smilax aspera* are surely based on the chemical composition of these two plants, for they have esteroidic and saponine components [59] that favor those sorts of viscous formulations. Table A1 also shows the infusions that could be prepared with the autochthonous elements of the territory. In the case of chamomiles, which are recommended for their stomachic and digestive properties, the different species (*Chamaemelum nobile*, *Chamaemelum fuscatum*, *Matricaria chamomilla*, and *Heychrysum stoechas*) provide different qualities and let the consumer choose according to their individual preferences. This is related to the chemical profile of the essential oil, which has been characterized by chromatography [60,61]. Other digestive genera include *Mentha*, *Origanum*, *Sideritis*, and *Thymus* [62]. The antiinflamatory profile of *Phlomis* [63] and *Malva* [64] or the sedative of other cases such as *Melissa*, *Hypericum*, or *Crataegus* [64] are very interesting from the nutraceutic point of view.

The Confectionery and Preserves groups have discrete representations (see Figure 3), which do not detract from the importance of particular species. The sweetening power of *Celtis australis* would be worth investigating for its improvement and exploitation of low-calorie programs, which are in the interest of the overweight public [65]. Others can be objects of creativity, such as ice creams, sorbets, or marmalades made out of the mature fruits of *Myrtus communis*, which have excellent antioxidant properties [66]. The brines of *Sedum album* (which is very rich in flavonol glycosides [67]) or the pickles of *Portulaca oleracea* (which have large amounts of ω-3 and ω-6 fatty acids [68] all represent really high gourmet challenges.

We must finally emphasize the importance of the Sauces group, since the most abundant list of species corresponds to those used as condiments. They are plants with essential oils, which are

well-known in popular gastronomy and mostly belong to the *Lamiaceae* and *Apiaceae* families. In Table A1, these are assigned to number 21. However, we must emphasize them over the relevance of natural rennets such as *Scolymus hispanicus*, *Cynara cardunculus*, and *Cynara humilis*, because they open many possibilities of introducing new taste sensations and presentations. Something similar can be said about a potential flour use of *Elymus repens* or *Scirpoides holoschoenus*, [69] based in traditional knowledge about vegetal biodiversity. The latter is another food technology and scientific challenge, because the starchy rhizome is very rich in resveratrol and other antioxidant molecules that should be better studied for its best utilization.

## 5. Conclusions

The envisaged promotion of local products throughout environmentally sustainable techniques further contributes to environmental protection. The valorization of traditional foods and recipes is necessary in order to simultaneously preserve the local agro-biodiversity, sustainability, and productive systems. In this context, it is becoming important to address the consumer folk knowledge toward innovative applications. This strategy should represent a valid tool for the promotion of socioeconomic development, the enhancement of territories, and biodiversity preservation.

**Author Contributions:** Conceptualization, T.R.-T.; Methodology, J.B.-S.; Investigation, J.B.-S., L.G.-G.; Resources, J.L.-M.; Data Curation, L.G.-G.; Writing-Original Draft Preparation, T.R.-T.; Writing-Review and Editing, L.G.-G.; Validation, J.L.-M.; Visualization, L.G.-G., J.L.-M.; Supervision, T.R.-T.; Project Administration, J.B.-S.; Funding Acquisition, T.R.-T., J.L.-M.

**Funding:** This research was by Junta de Extremadura (Spain) and European Regional Development Fund, through grant number IB16003.

**Acknowledgments:** To Manuel Pardo de Santayana (Universidad Autónoma de Madrid) and the Research Group "Proyecto Inventario Español de Conocimientos Tradicionales Relativos a la Biodiversidad". To Pedro Escobar García (Natural History Museum, Vienna, Austria) and two anonymous referees for their helpful corrections in the draft versions of the manuscript.

**Conflicts of Interest:** The authors declare no conflict of interest.

## Appendix A.

**Table A1.** Catalog of edible plants present in the protected area " Sierra Grande de Hornachos".

| Family | Scientific Name | Vernacular Name | PP | Subc | Pr | CL F |
|---|---|---|---|---|---|---|
| Adoxaceae | *Sambucus nigra* L. | Saúco | I, Fr | A, C, G, H | a, b | 4, 24, 29, 30, 33, 37, 38, 40 |
| | *Viburnum tinus* L. | Durillo | L | I | b | 21 |
| Amaranthaceae | *Chenopodium album* L. | Cenizo | Wp, R | A, B | a, b | 1, 4, 6, 7, 8, 10, 11, 12, 22 |
| | *Chenopodium murale* L. | Cenizo | Wp | A | c | 7 |
| Amaryllidaceae | *Narcissus bulbocodium* L. * | Campanitas | B, F | B, K | b | n.d. |
| | *Allium ampeloprasum* L. | Ajo porro | Wp, B, Bp | B, I | a, b | 1, 4, 6, 7, 8, 11, 12, 13, 14, 15, 21 |
| Anacardiaceae | *Allium roseum* L. | Ajo porro | L, B | A, B, I | a, b | 2, 7, 10, 11, 21, 24 |
| | *Pistacia lentiscus* L. | Lentisco | Ap, Wp, L, Sv | G, I, K, L | b | 20, 21, 38 |
| | *Apium graveolens* L. | Apio | Wp | A | a, b | n.d. |
| | *Apium nodiflorum* (L.) Lag. | Berra | St, Bs | A | a, b | 2, 7, 8, 11 |
| Apiaceae | *Foeniculum vulgare* Mill. | Hinojo | L, Bs, I, St, Wp, S, Fr | A, G, H, I, K, L | a, b | 1, 4, 6, 7, 9, 10, 11, 13, 21, 37, 38, 40, 41, 42 |
| | *Scandix australis* L. | Quijones | Wp, Ap | A, G, H, I | a, b | 11, 21, 38, 40, 41 |
| | *Scandix pecten-veneris* L. | Alfileres | L, Fr | A | a, b | 11 |
| | *Thapsia villosa* L. * | Cañaheja | R, Wp | B, G, I | a, b | 21, 38 |

**Table A1.** *Cont.*

| Family | Scientific Name | Vernacular Name | PP | Subc | Pr | CL F |
|---|---|---|---|---|---|---|
| Apocynaceae | *Vinca difformis* Pourr. * | Alcandorea | F | K | b | n.d. |
| Araceae | *Arisarum simorrhinum* Dur. * | Candil | Tu | B | c | 2 |
| | *Arum italicum* Mill. * | Aro | Tu | B | a | 5, 10, 14 |
| Araliaceae | *Hedera helix* L. * | Hiedra | St | A | c | n.d. |
| Aristolochiaceae | *Aristolochia paucinervis* Pomel * | Candil | St | A | a | 11 |
| | *Aristolochia pistolochia* L. * | Candil | St | A | a | 11, 12 |
| Asparagaceae | *Ruscus aculeatus* L. | Rusco | Bs, St | A, I | a, b | 4, 21, 24 |
| | *Urginea maritima* (L.) Baker * | Cebolla | L | I | b | 20 |
| Asphodelaceae | *Asphodelus aestivus* Brot. | Cebollino | L, Bs, Tu | A, B | a | 11 |
| | *Asphodelus albus* Mill. | Gamón | L, Bs, Tu | A, B | a | 11 |
| | *Asphodelus fistulosus* L. | Cebollino | L, Bs, Tu | A, B | a | 11 |
| | *Asphodelus ramosus* L. | Gamón | L, Bs, Tu | A, B | a | 11 |
| Aspleniaceae | *Asplenium ceterach* L. | Doradilla | Wp | G | b | 38 |
| Asteraceae | *Achillea ageratum* L. | Árnica | Wp | A, H | c | 44 |
| | *Anacyclus clavatus* (Desf.) Pers. | Magarza | L, Bs | A | c | n.d. |
| | *Andryala integrifolia* L. | Árnica | L, St | A | a, b | 7 |
| | *Andryala laxiflora* DC. | Árnica | L, St | A | b | 7 |
| | *Andryala ragusina* L. | Ajonje | St, Sv | K | b | n.d. |
| | *Anthemis cotula* L. | Magarza | Wp | G | c | 38 |
| | *Bellis perennis* L. | Margarita | L, F | A, K | b | 7 |
| | *Calendula arvensis* L. | Caléndula | Bs, F | A | a, b | 11 |
| | *Carthamus lanatus* L. | Cardo | L, F | A, I | a, b | 7, 10, 11, 19 |
| | *Chamaemelum fuscatum* (Brot.) Vasc. | Magarza | F | H | a | 42 |
| | *Chamaemelum nobile* (L.) All. | Manzanilla | I, F | G, H | a, b | 38, 42 |
| | *Chondrilla juncea* L. | Ajonjera | L, St, Wp, Sv | A, K | a, b | 1, 7, 11 |
| | *Cichorium intybus* L. | Achicoria | L, Bs, R, F | A, H, K | a, b | 1, 7, 9, 11, 12, 13, 15, 43 |
| | *Crepis capillaris* (L.) Wallr. | Almirón | Wp | A | c | n.d. |
| | *Crepis foetida* L. | Almirón | | A | c | n.d. |
| | *Crepis vesicaria* subsp *haenseleri* (Boiss. ex DC.) P.D.Sell | Achicoria | L | A | a,b | 4, 7, 10, 11 |
| | *Cynara cardunculus* L. | Cardo | L, St, F, Wp, Pn | A, I, K | a, b | 6, 7, 10, 11, 15, 19, 24, 33 |
| | Cynara humilis L. | Alcachofa | I, F | A, L | a, b | 19 |
| | *Dittrichia graveolens* (L.) Greuter | Hierba matapulgas | Ap | I | a | 21 |
| | *Dittrichia viscosa* (L.) Greuter | Olivarda fina | I | G | b | 38 |
| | *Galactites tomentosa* Moench | Cardo | St | A | c | n.d. |
| | *Helichrysum stoechas* (L.) Moech | Manzanilla | Pe, F, Ap | G, H, L | a | 38, 43 |
| | *Mantisalca salmantica* (L.) Briq. & Cavillier | Escoba | L, St | A | a, b | 7, 10, 11, 12 |
| | *Matricaria chamomilla* L. | Manzanilla | I, F | G, H | a, b | 40, 42 |
| | *Reichardia intermedia* (Schultz Bip) Samp. | Lechuguilla | | A | a, b | 7, 13, 15, 24 |
| | *Scolymus hispanicus* L. | Cardillo | L, St, R, F | A, I | a | 4, 11, 12, 13, 15, 19, 20, 21, 32 |
| | *Scorzonera angustifolia* L. | Teta de vaca | L, Pf, F | A, K | a, b | 7, 11 |
| | *Scorzonera hispanica* L. | Alcarcionera | Bl, R | K | b | |
| | *Scorzonera laciniata* L. | Berbaja | L, F, R, St, F | A, B, K | a, b | 4, 7, 11, 13 |
| | *Silybum marianum* (L.) Gaertner | Cardo mariano | Wp, S, Fr | A, I, K | a, b | 7, 10, 11, 13, 15, 19 |
| | *Sonchus asper* (L.) Hill | Cerraja | L | A | a, b | 6, 7, 9, 13, 15 |
| | *Sonchus oleraceus* L. | Cerraja | L, Bs, R | A,B | a, b | 2, 4, 5, 6, 7, 9, 10, 11, 12, 13, 14, 15, 24 |
| | *Sonchus tenerrimus* L. | Cerraja | L | A | a, b | 5, 7 |
| | *Tragopogon porrifolius* L. | Teta de vaca | L, I, R | A, K | a, b | 7, 11 |
| | *Urospermum picroides* (L.) Scop. ex F.W. Schmidt | Cerrajón | L, Bs, R | A, B, H | a, b | 7, 9, 13 ,15 |
| | *Xanthium spinosum* L. | Cadillo | Wp | G | c | 38 |
| Boraginaceae | *Borago officinalis* L. | Borraja | L, St, F | A, K | a, b | 4, 7, 11, 12, 15, 24 |
| Brassicaceae | *Capsella bursa-pastoris* (L.) Medik. | Bolsa de pastor | L, Ap, Fr | A, G, K | a, b | 1, 12, 38 |
| | *Eruca vesicaria* (L.) Cav. | Oruga | L, Bs, S | A, D, I | a, b | 4, 7, 8, 10, 11, 15, 21 |
| | *Rapistrum rugosum* (L.) All. | Ravaniza | L | A | a | 6, 10, 11 |
| | *Sisymbrium irio* L. | Rabaniza | L | A | a | 11 |
| | *Sisymbrium orientale* L. | Tamarilla | | A | a, b | 6, 8 |

**Table A1.** *Cont.*

| Family | Scientific Name | Vernacular Name | PP | Subc | Pr | CL F |
|---|---|---|---|---|---|---|
| Campanulaceae | *Campanula rapunculus* L. | Vara de San José | Bs, R, Fr | A,B,G | a,b | 7,13,38 |
| Caprifoliaceae | *Lonicera etrusca* G. Santi | Madreselva | F | K | b | n.d. |
| | *Lonicera implexa* Aiton | Madreselva | F, Ap | G, K | b | 38 |
| | *Lonicera periclymenum* subsp *hispanica* (Boiss & Reuter) Nyman | Madreselva | I, F | G, K | c | 38 |
| | *Herniaria cinerea* DC. | Arenaria | Wp | G | c | 38 |
| | *Herniaria glabra* L. | Arenaria | Wp | G | c | 38 |
| Caryophyllaceae | *Herniaria lusitanica* Chaudhri | Arenaria | Wp | G | c | 38 |
| | *Herniaria scabrida* Boiss | Arenaria | Wp | G | c | 38 |
| | *Silene vulgaris* (Moench) Garcke | Colleja | L, St, R, Wp | A, B, L | a, b | 4, 6, 7, 9, 10, 11, 12, 15, 34 |
| | *Stellaria media* (L.) Vill. | Pamplina | L | A | a, b | 7, 11 |
| Cistaceae | *Cistus albidus* L. | Estepa | S | D | c | n.d. |
| | *Cistus ladanifer* L. | Jara | F, S, Sv | A, D, K | b | n.d. |
| Crassulaceae | *Sedum album* L. | Arroz | L | A, K | b | 33, 34 |
| | *Umbilicus gaditanus* Boiss. | Campanitas | | A | c | n.d. |
| | *Umbilicus rupestris* (Salisb.) Dandy | Ombligo de Venus | L | A | a, b | 7, 11 |
| Cucurbitaceae | *Bryonia dioica* Jacq. | Nueza | Bs, Z | A | a, b | 1, 4, 7, 8, 11, 12 |
| Cyperaceae | *Scirpoides holoschoenus* (L.) Soják | Junco | S, Bl | E, K | a, b | 2, 22 |
| Cytinaceae | *Cytinus hypocistis* (L.) L. | Colmenita | F | K | b | 13, 15 |
| Dennstaedtiaceae | *Pteridium aquilinum* (L.) Kuhn | Helecho | Rz, L | B, K | a, b | 5, 22 |
| Dioscoraceae | *Tamus communis* L. | Espárrago | Bs | A | a, b | 4, 11, 13, 15 |
| Equisetaceae | *Equisetum ramosissimum* Desf. | Cola de caballo | Bs, St | A, G | a, b | 15, 38 |
| Ericaceae | *Arbutus unedo* L. | Madroño | Fr, L, St, Ba | C, G, I, K | a, b | 21, 24, 29, 30, 37, 38, 41 |
| | *Bituminaria bituminosa* (L) Stirton | Angelota | F | K | b | n.d. |
| Fabaceae | *Lygos sphaerocarpa* (L.) Boiss | Retama | St, Bs, L | I, L | b | 20 |
| | *Medicago sativa* L. | Alfalfa | Bs, Wp | A, G, H | a | 1, 4, 6, 7, 12, 13, 15, 38, 43 |
| | *Onobrychis humilis* (L.) G. López | Carretilla | F | K | b | n.d. |
| | *Spartium junceum* L. | Retama | F | G | c | 38 |
| Fagaceae | *Quercus rotundifolia* Lam. | Encina | Fr | D, F, G, H | a, b | 5, 13, 17, 22, 24, 27, 38, 40, 43 |
| | *Quercus suber* L. | Alcornoque | Fr | D | a | 13 |
| Gentianaceae | *Centaurium erythraea* subsp *grandiflorum* (Biv) Melderis | Centaura | Wp | G | c | 38 |
| | *Centaurium maritimum* (L.) Fritsch | Centaura | Wp | G | c | 38 |
| | *Centaurium pulchellum* (Swartz) Druce | Centaura | Wp | G | c | 38 |
| Hypericaceae | *Hypericum perforatum* L. | Hipérico | F, Ap | G, H, K | a, b | 38, 42 |
| | *Calamintha nepeta* (L.) Savi | Hierba nieta | Wp, Ap, F | G, H, I, L | a | 21, 38, 43 |
| | *Marrubium vulgare* L. | Marrubio | F. Ap | G | c | 38 |
| | *Melissa officinalis* subsp *altissima* (Sibth & Sm) Arcangeli | Toronjil | Fr, Ap, L, F | G, H, I | a, b | 21, 37, 38, 42 |
| Lamiaceae | *Mentha pulegium* L. | Poleo | F, Ap, L | G, H, I | a | 21, 38, 42 |
| | *Origanum vulgare* subsp. *vires* (Hoffmanns. & Link) Bonnier & Layens | Orégano | Wp, Ap, F | G, H, I | a, b | 21, 37, 38, 42 |
| | *Phlomis lychnitis* L. | Candilera | L, F | H, K | a, b | 42, 43 |
| | *Phlomis purpurea* L. | Matagallos | L | K | b | |
| | *Rosmarinus officinalis* L. | Romero | Bs, L, St, F | G, I, K | b | 17, 18, 21, 37, 38 |
| | *Sideritis hirsuta* L. | Rabo de gato | F, Ap | G, H | a | 38, 42 |
| | *Thymus mastichina* (L.) L. | Mejorana | F, Ap | G, I | b | 21, 38 |
| Malvaceae | *Malva sylvestris* L. | Malva | L, P, F, Fr | A, G, H, K | a, b | 7, 11, 15, 38, 40, 42 |
| Myrtaceae | *Myrtus communis* L. | Arrayán | Fr, F, Ap, L | C, G, H, I | a, b | 20, 21, 25, 30, 37, 38 |
| Oleaceae | *Olea europaea* L. | Acebuche | Fr | D | b | 17, 33 |
| | *Phillyrea angustifolia* L. | Labiérnago | Wp | I | a | 20 |

**Table A1.** *Cont.*

| Family | Scientific Name | Vernacular Name | PP | Subc | Pr | CL F |
|---|---|---|---|---|---|---|
| Oxalidaceae | *Oxalis corniculata* L. | Trébol | L | A | c | n.d. |
| | *Papaver dubium* L. | Amapola | L, P | A | a, b | 6, 7, 27 |
| | *Papaver hybridum* L. | Amapola | L, St | A | a, b | 7, 10, 11 |
| Papaveraceae | *Papaver pinnatifidum* Moris | Amapola | L | A | a, b | 6, 7, 27 |
| | *Papaver rhoeas* var. *rhoeas* L. | Amapola | L, St, P, F, S | A, G, I | a, b | 4, 7, 8, 10, 11, 12, 13, 15, 21, 38 |
| | *Roemeria hybrida* (L.) DC. | Amapola morada | Bs | A | a, b | 4, 7, 10, 12, 13 |
| Pinaceae | *Pinus pinea* L. | Pino piñonero | S, Fr | D, G, I | a, b | 11, 20, 21, 24, 34, 38 |
| Plantaginaceae | *Digitalis purpurea* subsp *toletana* (Font Quer) Hinz * | Digital | F | K | b | n.d. |
| | *Digitalis thapsi* L. * | Biloria | F | K | b | n.d. |
| | *Veronica anagallis-aquatica* L. | Berro | L, St | A, K | a, b | 7 |
| Poaceae | *Arundo donax* L. | Caña | St, Bl | I, K | b | 20, 21 |
| | *Cynodon dactylon* (L.) Pers. | Grama | Rz, Wp | B, G, H, K | a, b | 13, 22, 38, 40, 42 |
| | *Elymus repens* (L.) Gould | Grama | Rz | B | a | 4, 22 |
| Polygonaceae | *Rumex pulcher* L. | Romaza | L, Ap | A, H | a | 4, 6, 7, 10, 13, 15, 40, 43 |
| Portulacaceae | *Montia fontana* subsp. *amporitana* Sennen | Pamplina | L, St | A | b | 7 |
| | *Portulaca oleracea* L. | Verdolaga | L, St, Bs | A | a, b | 4, 7, 10, 11, 15, 34 |
| Rosaceae | *Agrimonia eupatoria* L. | Agrimonia | Wp | G | c | 38 |
| | *Crataegus monogyna* Jacq. | Espino albar | L, St, Bs, Fr, S | A, C, G, H | a, b | 30, 32, 41, 43 |
| | *Rubus ulmifolius* Schott | Zarzamora | L, Bs, F, Fr, A | A, C, G, H, K | a, b | 4, 13, 24, 30, 38, 40 |
| Rutaceae | *Ruta angustifolia* Pers. | Ruda | Wp, L, F, S | G, H, I | b | 4, 12, 17, 18, 21, 38, 40 |
| | *Ruta montana* L. | Ruda | Wp, L, F, S | G, H, I | b | 4, 12, 17, 18, 21, 38, 40 |
| Smilaceae | *Smilax aspera* var. *altissima* Moris & De Not | Zarzaparrilla | Bs, R | A, G, H | a, b | 12, 38, 40, 44 |
| Solanaceae | *Solanum nigrum* L. | Hierba mora | | C | b | n.d. |
| Ulmaceae | *Celtis australis* L. | Almez | Fr | C, G, J | b | 23, 38 |
| | *Ulmus minor* Miller | Olmo | L, Bs, F | A, K | b | n.d. |
| Urticaceae | *Urtica dioica* L. | Ortiga | Bs, Wp | A, G | a, b | 1, 3, 4, 7, 11, 12, 13, 38 |
| | *Urtica membranacea* Poiret | Ortiga | L, St | A | a | 3, 4, 7, 13 |
| | *Urtica urens* L. | Ortiga | L, St, Wp | A, G | a | 1, 4, 7, 10, 11, 12, 13, 15, 24, 38 |
| Valerianaceae | *Valerianella microcarpa* Desv. | Canónigo | L | A | c | 7 |

PP = Parts of the used plant (BL = Base of leaf/stem; Bp = Base of the pseudostem; Bs = Tender shoots; B = Bulb; Ba = Bark; F = Flowers; Fr = Fruits; L = Leaves; I = Inflorescence; Ap = Aerial part; Fp = Floral peduncle; Pn = "Penca"; P = Petals; Wp = Whole plant; R = Root; Rz = Rhizome; Sv = Sap/resin/exudate/latex; S = Seeds; St = Stems; Tu = Tubers; Z = tendrils). Subc = Subcategories of food uses proposed by the Spanish Inventory of Traditional Knowledge Relating to Biodiversity (A = Vegetables; B = Roots, bulbs, tubers, and rhizomes; C = Sweet fruits; D = Dry and oleaginosus fruits; E = Cereals and pseudocereals; F = Fat; G = Alcoholic drinks; H = Non-alcoholic drinks; I = Condiments; J = Sugars and sweeteners; K = Candies and chewing; L = Other food uses); Pr = Preparation (a = cooked, b = raw, c = unknown); Cl F = Classification of foods according to Bertrand (2015) (1 = Soups; 2 = "Gofio" (= roasted cereal stirred into liquid); 3 = Broths and purees; 4 = Omelettes; 5 = Cakes; 6 = Rices; 7 = Salads; 8 = Cold vegetable soups; 9 = Pies/Patties; 10 = Potages; 11 = Stews; 12 = Scrambled; 13 = Sautéed/boiled/toasted; 14 = Milky; 15 = Fried/breaded; 16 = Sauces; 17 = Oil; 18 = Vinegar; 19 = Renner; 20 = Preservative; 21 = Dressings/condiments; 22 = Flour; 23 = Sugar; 24 = Desserts; 25 = Ice creams; 26 = Sorbets; 27 = Cakes/Cakes; 28 = Sweet; 29 = Jams; 30 = Marmalades; 31 = Jellies; 32 = To the natural; 33 = Brine; 34 = Pickles; 35 = In syrup; 36 = In alcohol; 37 = Wines; 38 = Liquors; 39 = Syrups for cocktails; 40 = Juices; 41 = Brandies; 42 = Infusions; 43 = Coffee/tea; 44 = Diluted syrups). * Toxic plants. n.d. = no data.

## Appendix B.

**Box A1.** Vegetables (85 taxa).

The most common:

Asteraceae (30.59%), Liliaceae (9.41%), and Apiaceae (8.24%).

Part consumed:

Leaves (65.88%), followed by the tender shoots (35.29%) or the stems (24.71%). Other parts as the flowers or fruits are less used (10.59%, 5.88%). More seldom are tendrils, (2.35%), roots and bulbs, petals, or specifically the floral receptacle or peduncle (1.18%).

Consumption mode:

Five peculiar cases—*Andryala laxiflora, Bellis perennis, Crataegus monogyna, Montia fontana* subsp. *Amporitana,* and *Sedum album*—are exclusively consumed in crude for brine, entertainment, or salads.

The largest group of useful vegetable plants (50 species) is composed by those than can be consumed either raw or cooked in omelettes, stews, purées, soups, rice, and pastries; regarding their preparation, they can be fried, scrambled, or boiled. These are: *Allium roseum, Andryala integrifolia, Apium graveolens, Apium nodiflorum, Bryonia dioica, Calendula arvensis* subsp. *arvensis, Calendula arvensis* subsp. *macroptera, Campanula rapunculus, Carthamus lanatus, Chenopodium album, Chondrilla juncea, Cichorium intybus, Crepis vesicaria* subsp. *haenseleri, Cynara cardunculus* subsp. *cardunculus, Equisetum ramosissimum, Eruca vesicaria, Foeniculum vulgare, Foeniculum vulgare* subsp. *piperitum, Malva sylvestris, Mantisalca salmantica, Papaver dubium, Papaver hybridum, Papaver pinnatifidum, Papaver rhoeas* var. *rhoeas, Portulaca oleracea, Portulaca oleracea* subsp. *granulato stellulata, Reichardia intermedia, Roemeria hybrida, Rubus ulmifolius, Sambucus nigra, Scandix australis* subsp. *australis, Scandix australis* subsp. *microcarpa, Scandix pecten-veneris, Scorzonera angustifolia, Scorzonera laciniata, Silene vulgaris, Silybum marianum, Sisymbrium orientale, Smilax aspera* var. *altissima, Smilax aspera* var. *aspera, Sonchus asper, Sonchus oleraceus, Sonchus tenerrimus, Stellaria media, Tamus communis, Tragopogon porrifolius, Umbilicus rupestris, Urospermus picroides, Urtica dioica*, and *Veronica anagallis-aquatica*.

Eighteen species have been popularly consumed stewed: *Aristolochia paucinervis* and *Aristolochia pistolochia* (whose fresh stems are used to make a stew in which they are fried with oil and garlic, salt, bread crumbs, paprika, and eggs), *Asphodelus aestivus, Asphodelus albus* subsp. *albus, Asphodelus albus* subsp. *villarsii, Asphodelus fistulosus, Asphodelus ramosus, Borago officinalis* (although the flowers may be eaten raw), *Capsella bursa-pastoris, Cynara humilis* (from which the inflorescences are consumed), *Medicago sativa, Rapistrum rugosum, Rumex pulcher, Ruscus aculeatus* (in omelettes), *Scolymus hispanicus* subsp. *hispanicus, Sisymbrium irio, Urtica membranacea*, and *Urtica urens* (in salad after being scalded or cooked).

**Box A2.** Roots, bulbs, tubers, and rhizomes (21 taxa).

---

**Sweet fruits (seven taxa)**

Families:

Belonging to Rosaceae and five more families: Caprifoliaceae, Ericaceae, Myrtaceae, Solanaceae, and Ulmaceae.

Part consumed:

Fruits

Consumption mode:

*Arbutus unedo, Crataegus monogyna, Myrtus communis,* and *Rubus ulmifolius* have been used both for direct consumption or as raisins, as for elaborations such as liquors, jams, ice cream, and various desserts.

In the case of *Sambucus nigra*, although in some areas it was said that its fruits were poisonous, once ripe, they have been used to make jams.

Only in two cases—*Celtis australis* and Solanum *nigrum*—is the raw consumption of fruits mentioned as a treat.

**Dry and oleaginous fruits (eight taxa)**

Families:

Fagaceae, Oleaceae, Pinaceae, Brassicaceae, and Cistaceae belong to this group.

Part consumed:

Seeds and fruits

Consumption mode:

For the species *Pinus pinea* and *Quercus rotundifolia*, seeds can be eaten raw, but they are toasted, roasted, or cooked to make them more palatable. Regarding Quercus suber, its oak is eaten cooked, and it has stringent properties.

For the species *Cistus ladanifer*, *Eruca vesicaria,* and *Olea europaea,* raw seeds or fruits are consumed directly from the plant or after processing, such as in brines or oils.

Finally, we have reported the mention of elaborated *Cistus albidus* seeds as being very appetizing.

**Cereals and pseudocereals; legumes; fat (two taxa)**

Families:

Cyperacerae and Fagaceae.

Part consumed:

Seeds and fruits

Consumption mode:

*Scirpoides holoschoenus,* whose seeds can form a possible base for the preparation of flours.

*Quercus rotundifolia,* which takes advantage of the food fats in their fruits, extracting them by cooking.

---

**Box A3.** Alcoholic drinks (58 taxa).

---

The most common:

Lamiaceae (13.79%), Asteraceae (10.34%), Caryophyllaceae (10.34%), Apiaceae (8.62%), Gentianaceae (6.90%), and 20 more families.

Part consumed:

Flowers (25.86%) or the fruit/seeds (15.52%) of the whole of the aerial part (51.79%).

Consumption mode:

All are used as ingredients in the elaboration of liquors.

These species are: *Agrimonia eupatoria* subsp. *eupatoria, Anthemis cotula, Arbutus unedo, Asplenium ceterach, Calamintha nepeta* subsp. *nepeta = Satureja calamintha, Campanula rapunculus, Capsella bursa-pastoris, Celtis australis, Centaurium erythraea* subsp. *grandiflorum, Centaurium erythraea* subsp. *erythraea, Centaurium maritimum, Centaurium pulchellum, Chamaemelum nobile, Crataegus monogyna, Cynodon dactylon* var. *dactylon, Cynodon dactylon* var. *villosum, Dittrichia viscosa, Equisetum ramosissimum, Foeniculum vulgare, Foeniculum vulgare* subsp. *piperitum, Helichrysum stoechas, Herniaria cinerea, Herniaria glabra, Herniaria lusitanica, Herniaria lusitanica* var. *gaditana, Herniaria scabrida, Herniaria scabrida* subsp. *guadarramica, Hypericum perforatum, Lonicera implexa, Lonicera peryclimenum* subsp. *hispanica, Malva sylvestris, Marrubium vulgare, Matricaria chamomilla, Medicago sativa, Melissa officinalis* subsp. *altissima, Mentha pulegium, Myrtus communis, Origanum vulgare* subsp. *virens = Origanum virens, Papaver rhoeas* var. *rhoeas, Pinus pinea, Pistacia lentiscus, Quercus rotundifolia, Rosmarinus officinalis, Rubus ulmifolius, Ruta angustifolia, Ruta montana, Sambucus nigra, Scandix australis* subsp. *australis, Scandix australis* subsp. *microcarpa, Sideritis hirsuta, Smilax aspera* var. *altissima, Smilax aspera* var. *aspera, Spartium junceum, Thapsia villosa, Thymus mastichina, Urtica dioica, Urtica urens,* and *Xanthium spinosum*.

**Box A4.** Non-alcoholic drinks (32 taxa).

The most common:
Mostly Asteraceae (21.88%), Lamiaceae (18.75% ) and Apiaceae (12.50%).
Part consumed:
It is prepared with the plant or its flowers (90.63%), and less frequently (9.38%) with the fruits or seeds.
Consumption mode:
Elaboration uses include by decoction, maceration, or infusions, and to a lesser extent for syrups or juices (as in *Sambucus nigra*).

Seven of these plants are considered medicinal: *Achillea ageratum, Foeniculum vulgare, Foeniculum vulgare* subsp. *piperitum, Matricaria chamomilla, Scandix australis* subsp. *australis, Scandix australis* subsp. *Microcarpa*, and *Sideritis hirsuta*. The rest (*Chamaemelum fuscatum, Chamaemelum nobile, Cynodon dactylon* var. *dactylon, Cynodon dactylon* var. *villosum, Hypericum perforatum, Malva sylvestris, Melissa officinalis* subsp. *altissima, Mentha pulegium, Myrtus communis, Origanum vulgare* subsp. *virens, Phlomis lychnitis, Rubus ulmifolius, Rumex pulcher, Ruta angustifolia, Ruta montana, Sambucus nigra, Smilax aspera* var. *Altissima*, and *Smilax aspera* var. *aspera*) *are associated with refreshing properties, and whose drinks were taken for the mere pleasure of tasting them.*

Finally, seven species have been cited as older everyday drinks: *Calamintha nepeta, Cichorium intybus, Crataegus monogyna, Helichrysum stoechas, Medicago sativa, Quercus rotundifolia,* and *Urospermus picroides.*

**Box A5.** Condiments (32 taxa).

The most common:
Lamiaceae (18.75%), Apiaceae (15.63%), and Asteraceae (15.63%).
Part consumed:
The plant (31.25%), the leaf (28.13%), the flower (21.88%), the fruit/seed (19.01%), the stem (15.63%), and less frequently, the bark or the bulb (3.13%).
Consumption mode:
Twenty-two of these plants are specially significant for the dressing of olives, stews, and meats: *Allium ampeloprasum, Allium roseum, Calamintha nepeta* subsp. *nepeta = Satureja calamintha, Dittrichia graveolens, Eruca vesicaria, Foeniculum vulgare, Foeniculum vulgare* subsp. *piperitum, Melissa officinalis* subsp. *altissima, Mentha pulegium, Myrtus communis, Origanum vulgare* subsp. *virens = Origanum virens, Papaver rhoeas* var. *rhoeas, Rosmarinus officinalis, Ruscus aculeatus, Ruta angustifolia, Ruta montana, Scandix australis* subsp. *australis, Scandix australis* subsp. *microcarpa, Scolymus hispanicus* subsp *hispanicus, Thapsia villosa, Thymus mastichina,* and *Viburnum tinus.*

Six species (*Arbutus unedo, Arundo donax, Lygos sphaerocarpa = Retama sphaerocarpa, Phillyrea angustifolia, Pinus pinea,* and *Urginea maritima*) are appreciated for their preservative properties, and are used to maintain the quality of game meats, healing hams, olives, and cheeses.

Only in the case of *Pistacia lentiscus* do known uses include it both as a seasoning and as a preservative for olives and game meat. Finally, the flowers and fruits of *Carthamus lanatus, Cynara cardunculus* subsp. *Cardunculus,* and *Silybum marianum*, can be used for curdling milk and making cheeses.

**Box A6.** Sugars and sweeteners (one taxon).

Family:
Ulmaceae.
Part consumed:
Fruit.
Consumption mode:
*Celtis australis, Ulmaceae*, whose ripe fruits were boiled to be used as a sugar substitute in times of scarcity.

**Box A7.** Candies and chewing (43 taxa).

The most common:

Asteraceae (23.26%) and 26 more families.

Part consumed:

The parts of the plants of this group that have been consumed as sweets and masticatories are very diverse, from the flowers (48.84%), leaves (9.30%), stems (6.98%), fruits (6.98%), and roots/rhizome (11.63%), to the sap, latex (2.33%), or the gills (2.33%).

Consumption mode:

Except in specific cases in which the original product underwent a slight transformation (for example in the production of gum with the sap of Pistacia lentiscus), consumption was directly from the plant, and without subjecting them to any alteration of their properties and natural state.

The most appreciated part is the flowers, since in the case of 21 plants, this is the part that is most consumed (*Bellis perennis, Bituminaria bituminosa, Borago officinalis, Cichorium intybus, Cytinus hypocistis, Cytinus hypocistis* subsp *macranthus, Digitalis purpurea* subsp *toletana, Digitalis thapsi, Hypericum perforatum, Lonicera etrusca, Lonicera implexa, Lonicera peryclimenum* subsp. *hispanica, Narcissus bulbocodium, Onobrychis humilis, Onobrychis humilis* var. *glabrescens, Phlomis lychnitis, Rosmarinus officinalis, Scorzonera angustifolia, Scorzonera laciniata, Ulmus minor,* and *Vinca difformis*). The resting 22 species of this group are: *Andryala ragusina, Arbutus unedo, Arundo donax, Capsella bursa-pastoris, Chondrilla juncea, Cistus ladanifer, Cynara cardunculus* subsp. *cardunculus, Cynodon dactylon* var. *dactylon, Cynodon dactylon* var. *villosum, Foeniculum vulgare, Foeniculum vulgare* subsp. *piperitum, Malva sylvestris, Phlomis purpurea, Pistacia lentiscus, Pteridium aquilinum, Rubus ulmifolius, Scirpoides holoschoenus =* *Scirpus holochoenus, Scorzonera hispanica, Sedum album, Silybum marianum, Tragopogon porrifolius,* and *Veronica anagallis-aquatica.*

**Box A8.** Other food uses (nine taxa).

Families:

Diverse and singular (Anacardiaceae, Apiaceae, Asteraceae, Caryophyllaceae, Fabaceae, and Lamiaceae).

Part consumed:

In most cases (66.67%), the whole plant is applied.

Consumption mode:

*Calamintha nepeta* and *Helichrysum stoechas* improve the quality of wine; the florets of *Cynara humilis* can be used as curd milk for obtaining cheese; *Silene vulgaris* can be used to clean the cheeses; *Foeniculum vulgare* and/or *Foeniculum vulgare* subsp *piperitum* can be used as cooking water to wash the pig guts (in the slaughters) to disinfect them and neutralize the smell; the tender shoots and buds of *Lygos sphaerocarpa* and *Pistacia lentiscus* can be used to purify water tanks. Finally, *Matricaria chamomilla* removes the bad taste after drinking a bitter almond.

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
