# Peer review of "Wild Plants Potentially Used in Human Food in the Protected Area "Sierra Grande de Hornachos" of Extremadura (Spain)"

_sustainability, doi:10.3390/su11020456_

Reviewer 1 Report

1.        The manuscript is of merit and interest as it provides significant and useful data about edible plant resources traditionally used in a protected are of the SW Iberian Peninsula within Mediterranean culture. Moreover, the authors highlight relevant cultural heritage, important skills on the brink of disappearing, and bio patrimony, e.g. providing a catalogue of the edible local flora (145 species) and specific uses which is linked to a Natura 2000 network site.

2.        The theme is scientifically sound; the title and abstract introduce readers to general context and main objectives of the paper. However, the study area and the research context are not satisfactorily described, considering target readers of open access journals from MDPI.

3.        The introduction section is a bit dense writing and puzzled. It would be better if authors were able to create subsections considering the main topics addressed:

a.        Plant resources global availability and potential use;

b.       Combined conservation trends concerning species, uses and knowledge, focusing Natura 200 network;

c.        The particular case of the protected site known as Sierra Grande Hornados, pointing out constrains and social conflicts, as well as public awareness of the importance (.e.g. biological, ecological, socioeconomic and cultural relevance) of these protected sites;

d.       Local traditional knowledge on food plants and resource management, i.e. at Sierra Grande Hornados;

e.        New trends and development opportunities: wild gathering, agro-tourism, gourmet foods, local memories;

f.         General and specific objectives of the paper.

4.        The methods seem appropriate and basically consisted on a search performed in two databases: the IECTB Inventory and the Herbarium UNEX. However, this section needs a better description. For instance, the authors should have in mind that many readers will not be familiar with the IECTB Project, the Herbarium Unex or the Anthos project. Moreover it is also important to provide some language review:

Some examples:

Lines: 90-91, it should be: The database of the Spanish Inventory of Traditional Knowledge Relating to Biodiversity (IECTB) comprising of 3156 records (taxa), was taken as a basis [18–22].

Lines: 97-98, it should be: Hornachos (1301) plants were looked for in IECTB Inventory official database and 834 species matched. (Figure 1)

Line: 100, it should be: The 834 Hornachos useful plants included in IECTB.

Considering botanical nomenclature and taxonomical classification, I think it’s worth mentioning which systems the authors have followed in the paper for naming the species and botanical families, e.g. Anthos, Flora Iberica, APG, other…

Consistency is important: for instance Asparagaceae is not a botanical family considered in Anthos or Flora Iberica; otherwise, Umbelliferae, along the test, is not considered in APG system or the Plant List.

5.        Results section provides important and extensive data on the main topic of the paper but some information is codified, shortly presented, sending to a Figure or Table, which does not make easy for readers to understand the main data.

Results section denotes that authors’ approaches were not sufficiently detailed and explained before in methods section. Maybe methodological issues should be better addressed in methods section, specifically summarizing the IECTB subcategories of food and Bertrand culinary classification.

Moreover, considering that we have a very comprehensive Table 1 that introduces to the list of the plants and nomenclature, global results about the food use categories could be structured in a short and more comprehensive Table which might be better discussed along the text. Instead of complex coding and long list of names I suggest general information about significant uses and emblematic/cultural identity species.

I think the structure of the results section could be improved in order to emphasize the documented data. Figure 2 is not analyzed easily. I suggest using this figure with fewer values, just providing some examples highlighting the most significant families or species.

6.        Discussion and conclusions focus the major findings and reinforcing the importance of ethnobotanical studies to highlight local knowledge and significant culture/patrimony. Moreover, the potential of such information is also explored.

7.        The paper is a bit dense writing, not easy to understand on first reading because sometimes it seems the English translation is not accurate. Moreover some paragraphs are longer than it is necessary and used in English. Then, some sentences and paragraphs/ideas result a bit difficult to understand (e.g. lines 65-75; 207-210; 265-268).

I would suggest some important language revision in order to improve general comprehension and to highlight such relevant findings and testimony.

8.        Please check font size and format. It seems some lines have letters in bigger size, for instance lines 274. It might be due to pdf format?

9.        Some terms used are not the most appropriate in English writing.

10.     The literature cited seems pertinent but the final list needs some revision according to journal guidelines and to improve consistency among all references.

Author Response

1.        The manuscript is of merit and interest as it provides significant and useful data about edible plant resources traditionally used in a protected are of the SW Iberian Peninsula within Mediterranean culture. Moreover, the authors highlight relevant cultural heritage, important skills on the brink of disappearing, and bio patrimony, e.g. providing a catalogue of the edible local flora (145 species) and specific uses which is linked to a Natura 2000 network site. Thank you.

2.        The theme is scientifically sound; the title and abstract introduce readers to general context and main objectives of the paper. However, the study area and the research context are not satisfactorily described, considering target readers of open access journals from MDPI. Ok. Introduction has been modified.

3.        The introduction section is a bit dense writing and puzzled. It would be better if authors were able to create subsections considering the main topics addressed: Done. Ok.

a.        Plant resources global availability and potential use;

b.       Combined conservation trends concerning species, uses and knowledge, focusing Natura 200 network;

c.        The particular case of the protected site known as Sierra Grande Hornados, pointing out constrains and social conflicts, as well as public awareness of the importance (.e.g. biological, ecological, socioeconomic and cultural relevance) of these protected sites;

d.       Local traditional knowledge on food plants and resource management, i.e. at Sierra Grande Hornados;

e.        New trends and development opportunities: wild gathering, agro-tourism, gourmet foods, local memories;

f.         General and specific objectives of the paper.

4.        The methods seem appropriate and basically consisted on a search performed in two databases: the IECTB Inventory and the Herbarium UNEX. However, this section needs a better description. For instance, the authors should have in mind that many readers will not be familiar with the IECTB Project, the Herbarium Unex or the Anthos project. Moreover it is also important to provide some language review: Ok.

Some examples:

Lines: 90-91, it should be: The database of the Spanish Inventory of Traditional Knowledge Relating to Biodiversity (IECTB) comprising of 3156 records (taxa), was taken as a basis [18–22]. Ok.

Lines: 97-98, it should be: Hornachos (1301) plants were looked for in IECTB Inventory official database and 834 species matched. (Figure 1).

Line: 100, it should be: The 834 Hornachos useful plants included in IECTB. Ok.

Considering botanical nomenclature and taxonomical classification, I think it’s worth mentioning which systems the authors have followed in the paper for naming the species and botanical families, e.g. Anthos, Flora Iberica, APG, other…

Consistency is important: for instance Asparagaceae is not a botanical family considered in Anthos or Flora Iberica; otherwise, Umbelliferae, along the test, is not considered in APG system or the Plant List. Ok.

5.        Results section provides important and extensive data on the main topic of the paper but some information is codified, shortly presented, sending to a Figure or Table, which does not make easy for readers to understand the main data. You are right.

Results section denotes that authors’ approaches were not sufficiently detailed and explained before in methods section. Maybe methodological issues should be better addressed in methods section, specifically summarizing the IECTB subcategories of food and Bertrand culinary classification. This is true. Done. It has been included.

Moreover, considering that we have a very comprehensive Table 1 that introduces to the list of the plants and nomenclature, global results about the food use categories could be structured in a short and more comprehensive Table which might be better discussed along the text. Instead of complex coding and long list of names I suggest general information about significant uses and emblematic/cultural identity species. Yes. It is understood. Ok. Done.

I think the structure of the results section could be improved in order to emphasize the documented data. Figure 2 is not analyzed easily. I suggest using this figure with fewer values, just providing some examples highlighting the most significant families or species.

6.        Discussion and conclusions focus the major findings and reinforcing the importance of ethnobotanical studies to highlight local knowledge and significant culture/patrimony. Moreover, the potential of such information is also explored. Thank you.

7.        The paper is a bit dense writing, not easy to understand on first reading because sometimes it seems the English translation is not accurate. Moreover some paragraphs are longer than it is necessary and used in English. Then, some sentences and paragraphs/ideas result a bit difficult to understand (e.g. lines 65-75; 207-210; 265-268). Ok. It should be been set to MPDI translators as we exposed in the first paragraph of our coverletter.

I would suggest some important language revision in order to improve general comprehension and to highlight such relevant findings and testimony.

8.        Please check font size and format. It seems some lines have letters in bigger size, for instance lines 274. It might be due to pdf format? Yes.

9.        Some terms used are not the most appropriate in English writing. Ok.

10.     The literature cited seems pertinent but the final list needs some revision according to journal guidelines and to improve consistency among all references. OK. (Something wrong happened with Mendeley in the version we sent. Please excuse us.) We hope now it is OK.

Reviewer 2 Report

1.                For me the paper is a bit juggling with data. The authors analyze the potential of using wild plants of an area based on the information on their uses from Spanish ethnobotanical databases. However they do not discuss:

a.             Uses of these plants outside Spain.

b.            Field studies from Hornachos (what actual people use around there).

c.             Abundance of these plants which is of utmost importance if we want to use them for food.

2.                If the authors want to expand the paper I propose reading:

Jug-Dujakovic, Marija; Luczaj, Lukasz. The contribution of Josip Bakic's research to the study of wild edible plants of the Adriatic coast: A military project with ethnobotanical and anthropological implications. Slovensky Narodopis, 2016, 64.2: 158 [available as open pdf from publisher].

Which describes a series of studies with a similar paper done by Bakić for the Yugoslav Army (also a Mediterranean area).

3.                Qinlin should be spelled Qinling.

Author Response

1.                For me the paper is a bit juggling with data. The authors analyze the potential of using wild plants of an area based on the information on their uses from Spanish ethnobotanical databases. (and on the information of the wild flora of the area which had not been published, nor collected to Herbarium, not studied to date, despite to be a Protected Natura 2000 Area. We understand something must be misunderstood in the Introduction, or underexplained, and needs to be modified) However they do not discuss:

a.             Uses of these plants outside Spain. Because they are not most relevant for the main objective of the paper.

b.             Field studies from Hornachos (what actual people use around there). Because the paper (and the project that is financially supporting it), is mainly focused on giving an use to the wild flora actually existing in the area. We are not so interested on knowing what local people nowadays still use, but what can the flora be used for.. The project is more oriented to social innovation than to cultural history or ethnography or anthropology cultural.

c.             Abundance of these plants which is of utmost importance if we want to use them for food. All the plants listed are actually abundant as they have been  collected at the fields. In 2016/2017 by our team in the context the Project. Voucher hebariums Sheets are deposited in UNEX.

2.                If the authors want to expand the paper I propose reading:

Jug-Dujakovic, Marija; Luczaj, Lukasz. The contribution of Josip Bakic's research to the study of wild edible plants of the Adriatic coast: A military project with ethnobotanical and anthropological implications. Slovensky Narodopis, 2016, 64.2: 158 [available as open pdf from publisher].

Which describes a series of studies with a similar paper done by Bakić for the Yugoslav Army (also a Mediterranean area). Thank you very much. A great paper and contribution. We have read it with interest, and we have included a lot of its cites and reflexions in our Introduction (an discussion). It is really a singular contribution to ethnobotany in Europe. It must be designed as an incredible work. We are grateful for these author.

Even though we would like to expose that we hope that with our new version, the above mentioned concerns are clarified. Because we have included much more bibliography, and sometimes there are mentions of the uses of these plants outside Spain. Even though, that is not the relevant focus of our project. No yet is the use that actual people make in Hornachos. We will study that topic in the future. Now we are focused in comuunicate what are the possibilities of the Flora of this Natura 200 Territory for Development, Turism, etc. And  the abundance of all the plants cited in the text is insured in the area because it has been specifically the task of the first year of Phase of the Project we are actually carrying out there. This paper is associated to it.

3.                Qinlin should be spelled Qinling. Ok.

Reviewer 3 Report

1.        The matter of this paper is very interesting in the context of biodiversity and sustainability. The paper is suitable for publication after following revisions:

2.        The Material and Methods should be described with major details.

3.        The items in Table 1 should be divided in subgroups.

4.        The Figure 2 is not clear. The authors should improve it.

5.        The Discussion should be improved by mentioning recent references of elaboration of experimental recipes.

Author Response

1.        The matter of this paper is very interesting in the context of biodiversity and sustainability. The paper is suitable for publication after following revisions:

2.        The Material and Methods should be described with major details. Ok, done.

3.        The items in Table 1 should be divided in subgroups. Ok, done.

4.        The Figure 2 is not clear. The authors should improve it. Ok, done.

5.        The Discussion should be improved by mentioning recent references of elaboration of experimental recipes. Ok, done.

Reviewer 4 Report

The topic is suitable for publication in Sustainability.

Author Response

The topic is suitable for publication in Sustainability. Thank you

Round  2

Reviewer 1 Report

Thank you for reviewing your paper. This version will improve general comprehension, highlighting relevant findings and significant testimony.

Reviewer 2 Report

The authors failed to address the issue of edible uses of the local species which were recorded outside Spain.

The English is really very bad and difficult to understand.

283 Allium and Rapistrum are from completely different families